# Investigating the active chemical constituents and pharmacology of *Nanocnide lobata* in the treatment of burn and scald injuries

**Yanlin Zou**[1,2,3], **Cao Yu**[1,3], **Qian Huang**[1,3], **Xiaorong Tan**[1,3], **Xiaoyan Tan**[1,3], **Xiaolong Zhu**[1,3], **Dongyang Yi**[1,3]*, **Jingxin Mao**[2,4]*

**1** School of Pharmacy, Three Gorges Medical College, 404120, Chongqing, China, **2** School of Pharmacy, Chongqing Medical and Pharmaceutical College, 401331, Chongqing, China, **3** Chongqing Anti-tumor Natural Drug Engineering Technology Research Center, Three Gorges Medical College, 404120, Chongqing, China, **4** College of Pharmaceutical Sciences, Southwest University, 400715, Chongqing, China

* 1135754194@qq.com (DY); mmm518@163.com, maomao1985@email.swu.edu.cn (JM)

**Data Availability Statement:** All relevant data are within the manuscript and its Supporting information files.

## Abstract

### Objective

To identify the most effective fraction of *Nanocnide lobata* in the treatment of burn and scald injuries and determine its bioactive constituents.

### Methods

Chemical identification methods were used to analyze solutions extracted from *Nanocnide lobata* using petroleum ether, ethyl acetate, n-butanol using a variety of color reactions. The chemical constituents of the extracts were identified by ultra-performance liquid chromatography (UPLC)–mass spectrometry (MS). A total of 60 female mice were randomly divided into the following 6 groups: the petroleum ether extract-treated group; the ethyl acetate extract-treated group; the n-butanol extract-treated group; the model group; the control group; and the positive drug group. The burn/scald model was established using Stevenson's method. At 24 hours after modeling, 0.1 g of the corresponding ointment was evenly applied to the wound in each group. Mice in the model group did not undergo treatment, while those in the control group received 0.1 g of Vaseline. Wound characteristics, including color, secretions, hardness, and swelling, were observed and recorded. Photos were taken and the wound area calculated on the 1st, 5th, 8th, 12th, 15th, 18th and 21st days. Hematoxylin-eosin (HE) staining was utilized to observe the wound tissue of mice on the 7th, 14th, and 21st days. An enzyme-linked immunosorbent assay (ELISA) kit was used to measure the expression of tumor necrosis factor (TNF)-α, interleukin (IL)-10, vascular endothelial growth factor (VEGF) and transforming growth factor (TGF)-$\beta$1.

### Results

The chemical constituents of *Nanocnide lobata* mainly include volatile oils, coumarins, and lactones. UPLC–MS analysis revealed 39 main compounds in the *Nanocnide lobata* extract. Among them, ferulic acid, kaempferitrin, caffeic acid, and salicylic acid have been confirmed

**Funding:** This work was supported by Chongqing Municipal Education Commission Science and Technology Research Project—"Research on the active ingredients of Nanocnide lobata (snow medicine) in the treatment of burns and scald", NO. KJQN201802705; Chongqing Municipal Health and Family Planning Commission Traditional Chinese Medicine Science and Technology Project— "Pharmacodynamic observation and mechanism study of Nanocnide lobata (snow medicine) on burns and scald", NO. ZY201702136; Scientific research and seedling breeding project of Chongqing Medical Biotechnology Association, cmba2022kyym-zkxmQ0003; and 2022 scientific research project of Chongqing Medical and Pharmaceutical College, ygz2022104 respectively.

**Competing interests:** The authors declare that they have no conflict of interest.

to exhibit anti-inflammatory and antioxidant activity related to the treatment of burns and scalds. HE staining revealed a gradual decrease in the number of inflammatory cells and healing of the wounds with increasing time after *Nanocnide lobata* extract administration. Compared with the model group, the petroleum ether extract-treated group showed significant differences in the levels of TNF-α (161.67±4.93, 106.33±3.21, 77.67±4.04 pg/mL) and IL-10 (291.77±4.93, 185.09±9.54, 141.33±1.53 pg/mL) on the 7th, 14th, and 21st days; a significant difference in the content of TGF-$\beta$1 (75.68±3.06 pg/mL) on the 21st day; and a significant difference in the level of VEGF (266.67±4.73, 311.33±10.50 pg/mL) on the 7th and 14th days respectively.

## Conclusion

Petroleum ether *Nanocnide lobata* extract and the volatile oil compounds of *Nanocnide lobata* might be effective drugs in the treatment of burn and scald injuries, as they exhibited a protective effect on burns and scalds by reducing the expression of TNF-α, IL-10 and TGF-β1 and increasing the expression of VEGF. In addition, these compounds may also exert pharmacological effects that promote wound tissue repair, accelerate wound healing, and reduce scar tissue proliferation, inflammation and pain.

## 1. Introduction

Burn and scald injuries consist of damage to the skin or other body tissue caused by thermal radiation, which occurs when some or all of the cells of the skin or other tissue are damaged by hot liquids (scalds), solids (contact burns), or flames (flame burns) [1]. Other injuries to the skin or other body tissue caused by radiation, electrical current, friction, or exposure to chemicals are also considered burns [2]. Approximately 100 million people in the world suffer from burns of varying degrees every year, and the number of deaths caused by burns is second only to that caused by traffic accidents [3]. Severe burns and scalds cause both physical and mental harm, making it difficult for those affected to reintegrate into society [4]. In patients with large-scale burns and scalds, due to wound exudation and excessive water loss, the imbalance of body fluids and electrolytes is likely to cause fever, thereby reducing the physical fitness of the human body and affecting wound healing [5]. Additionally, long-term fever can easily lead to various complications and further endanger life. Burns and scalds can also cause serious damage to human skin, which may result in scars and disfigurement in severe cases [6].

*Nanocnide lobata* Wedd.N.pilosa Migo (snow medicine) is mainly distributed in shady and humid places in Chongqing, Sichuan, Hubei, Guangdong, Guangxi, Guizhou, Yunnan and other places in China. The family name of *Nanocnide lobata* is *Urticaceae* which less reports on the chemical components of it. It has been reported that the possible types of chemical components in *Nanocnide lobata* are organic acids, polysaccharides and glycosides, steroids or triterpenoids, flavonoids, coumarins, lactones, and volatile oils [7]. *Nanocnide lobata* is a traditional Chinese medicine (TCM) that is usually used to treat lung heat and cough, scrofula, hemoptysis, burns and scalds, carbuncles, bruises, snakebites, and traumatic bleeding [7,8]. *Nanocnide lobata* is commonly used to treat burns and scalds in Chinese folk medicine. However, there has been little modern clinical research on the pharmacology and efficacy of *Nanocnide lobata* in the treatment of burn and scald injuries [7,8]. In addition, there have been no systematic studies on the active fraction or chemical constituents of *Nanocnide lobata* in the treatment of burn and scald injuries.

The chemical composition of medicinal materials is the basis for their pharmacology and clinical efficacy. In the present experiment, petroleum ether, ethyl acetate and n-butanol were used to extract different active fractions of *Nanocnide lobata*. Chemical identification methods were utilized to investigate the chemical composition of each extracted fraction of *Nanocnide lobata* and provide a chemical foundation for further mechanistic research. In addition, active components in the treatment of burn and scald injuries were clarified.

## 2. Materials and methods

### 2.1 Materials

The original herbarium of *Nanocnide lobata* was kept in the 407 Natural medicinal chemistry Laboratory of the Scientific Research Center of Chongqing Three Gorges Medical College (No.20210503) in Wanzhou District, Chongqing, China. Specimens were gathered in April 2021 and identified as the aerial part of *Nanocnide lobata* Wedd., a plant of the Urticaceae family, by Professor Yi, School of Pharmacy, Chongqing Three Gorges Medical College. Approximately 760 g of dried *Nanocnide lobata* herbs was weighed and pulverized into primary powder. Then, 12 L (4 L×3) of 95% ethanol was added; the mixture was soaked for 48 h and then filtered with a 200-mesh filter cloth to obtain the filtrate. The filtrate was concentrated under reduced pressure to obtain 208.32 g of total extract at 40˚C and 0.1 MPa. *Nanocnide lobata* extraction was performed with ethanol and an appropriate amount of water using Han's method [9], with petroleum ether using Hacıbekiroğlu's method [10], with ethyl acetate using Afsar's method [11] and with n-butanol using Stoffers's method [12] 3 times at a ratio of 1:1. Finally, the mixture was concentrated under reduced pressure to obtain 15.83 g of petroleum ether extract, 2.54 g of ethyl acetate ester extract, and 13.58 g of n-butanol extract. Then, the petroleum ether extract, ethyl acetate ester extract, and n-butanol extract were each mixed with white Vaseline at a proportion of 85% white Vaseline and 15% extract to obtain ointments of suitable consistency for the further screening of active fractions.

### 2.2 Reagents

Anhydrous ethanol, 95% ethanol, petroleum ether (60~90˚C), n-butanol, ethyl acetate, ammonia water, potassium hydroxide, hydrochloric acid, sodium hydroxide, sulfuric acid, glacial acetic acid, acetic anhydride, chloroform, trichloromethane ferric chloride, sodium chloride, gelatin, vanillin, ninhydrin, copper sulfate, α-naphthol, copper sulfate, potassium sodium tartrate, bromophenol blue, bromocresol green, aluminum trichloride, magnesium powder, hydrogen oxide, boric acid, magnesium acetate, hydroxylamine hydrochloride, 3,5-dinitrobenzoic acid, 2,4,6-trinitrophenol, pyridine, sodium nitroferricyanide, iodine, potassium iodide, bismuth subnitrate, sodium silicotungstate, phosphomolybdic acid, potassium ferricyanide, 1% pentobarbital sodium, 4% paraformaldehyde universal tissue fixative, white petrolatum, "Jing wan hong" ointment (positive drug) and other reagents used to prepare chemical solutions were all analytically pure, and the water was purified water. Enzyme-linked immunosorbent assay (ELISA) test kits, including a mouse vascular endothelial growth factor (VEGF) ELISA kit, mouse transforming growth factor *β* 1 (TGF-*β*1) ELISA detection kit, mouse tumor necrosis factor α (TNF-α) ELISA test kit, and mouse interleukin-10 (IL-10) ELISA test kit, were purchased from Shanghai Future Industry Co., Ltd., and used for further study.

### 2.3 Experimental animals

Specific pathogen-free (SPF) female mice (weighing 20–30 g) supplied by Hunan Slike Jingda Laboratory Animal Co., Ltd., were used in the experiment. They were kept in separate cages

with 10 mice per cage in a clean-grade animal laboratory at constant temperature of 25±10˚C with a relative humidity of 40% to 70%. The breeding environment just kept at light-dark cycle was 12 h. Adapt to feeding for consecutive 3 days, fasting 12 h before the experiment, and drinking water freely.

## 2.4 Instruments

The following equipment was used in this study: vertical blast drying oven (Shanghai Jinwen Instrument Co., Ltd.); RE-5298 rotary evaporator (Shanghai Yarong Biochemical Instrument Factory); HLD-10002 electronic balance (Hangzhou Youheng Weighing Equipment Co., Ltd.); KH2200DB CNC Ultrasonic cleaner (Kunshan Hechuang Ultrasonic Instrument Co., Ltd.); HH-6 digital constant temperature water bath (Changzhou Yichen Instrument Manufacturing Co., Ltd.); SHZ-D (III) circulating water-type multipurpose vacuum pump (Henan Yuhua Instrument Co., Ltd.); BW-YLS-5Q desktop temperature control scald instrument (Shanghai Ruanlong Technology Development Co., Ltd.); HH-6 digital constant temperature water bath (Changzhou Yichen Instrument Manufacturing Co., Ltd.); DNM-9606 enzyme label analyzer (American Boten Instrument Co., Ltd.); 80–2 desktop centrifuge (Changzhou Jintan Liangyou Instrument Co., Ltd.); KD-BM biological tissue embedding machine (Zhejiang Jinhua Kedi Instrument Equipment Co., Ltd.); LEICA RM2235 slicer (Beijing Haonuo Technology Co., Ltd.); ultra-performance liquid chromatography (UPLC)–mass spectroscopy (MS) system (UPLC, Vanquish; MS, HFX, Thermo Scientific Co., Ltd.); Q Exactive HFX Hybrid Quadrupole Orbitrap mass spectrometer equipped with a heated electron spray ionization (ESI) source (Thermo Fisher Scientific Co., Ltd.); Q Exactive using Xcalibur 4.1 (Thermo Scientific Co., Ltd.); Progenesis QI software (Waters Corporation, Milford, USA); and Milli-Q water purification system (Millipore, Bedford, MA, USA).

## 2.5 Preliminary chemical identification tests

Preliminary chemical identification tests were carried out by a variety of color or precipitation reactions using indicator and chromogenic agents. Finally, the chemical composition of the *Nanocnide lobata* extracts were identified.

## 2.6 Extraction of bioactive ingredients from *Nanocnide lobata*

**2.6.1 Preparation of samples.** *Nanocnide lobata* powder was subjected to extraction by maceration with 95% ethanol overnight at room temperature for 3 consecutive days to yield the sample. Approximately 1 ml of the sample was added to 2 times the volume of the methanol-acetonitrile extraction solution (1:1, v/v), vortexed for 60 s, and sonicated for 30 min. After centrifugation (20 min, 12000 rpm, 4˚C), the supernatant was transferred to a clean plastic microtube. The sample was incubated for 1 h at -20˚C and then centrifuged at 12000 g at 4˚C for 10 min to remove the protein. The mixture was then centrifuged for 10 min (12000 g, 4˚C), and the supernatant was dried in a vacuum centrifuge. Subsequently, the sample was redissolved in 100 μl of 30% methyl cyanide (vol/vol) and transferred to an insert-equipped vial for analysis.

**2.6.2 UPLC–MS analyses.** Analysis of the bioactive compounds of *Nanocnide lobata* was carried out using a UPLC–MS system coupled to a Q Exactive HFX Hybrid Quadrupole Orbitrap mass spectrometer equipped with a heated ESI source utilizing the full-ms-ddMS2 MS acquisition method. The analytical conditions were as follows: UPLC: column, Waters HSS T3 (100×2.1 mm, 1.8 μm); column temperature, 40˚C; mobile phase: 0.1% formic acid aqueous solution as phase A and 0.1% formic acid acetonitrile as phase B; flow rate, 0.3 mL/min; injection volume, 2 μL; solvent system, water (0.1% acetic acid): acetonitrile (0.1% acetic acid); and

gradient program, 0 min, 0% phase B; 1 min, 0% phase B; 9 min, 95% phase B; 13 min, 95% phase B; 13 min, 0% phase B; 17 min, 0% phase B. The ESI source parameters were set as follows: spray voltage, -2.8 kV/3.0 kV; sheath gas pressure, 40 arb; aux gas pressure, 10 arb; sweep gas pressure, 0 arb; capillary temperature, 320°C; and aux gas heater temperature, 350°C. The raw MS data were acquired on the Q Exactive using Xcalibur 4.1 and processed using Progenesis QI.

### 2.7 Pharmaceutical trial validation

**2.7.1 Establishment of the scald/burn model and treatment of mice.** All mice were randomly divided into the following 6 groups, with 10 mice in each group: the petroleum extract-treated group; the ethyl acetate extract-treated group; the n-butanol extract-treated group; the positive drug group; the model group; and the control group. The scald/burn model was established following Stevenson's method [13]. After mice in each group were anesthetized by an intraperitoneal injection of 1% pentobarbital sodium (80 mg/kg), fur across approximately 16 cm$^2$ (4 cm× 4 cm) of the mouse back was removed with a depilator, and the small amount of remaining fur was removed with depilatory cream. A copper metal rod with a diameter of 1 cm was heated to 95°C and placed on the depilated area for 15 s, resulting in a shallow second-degree scald, with a wound size of approximately 150–200 mm$^2$. Twenty-four hours after modeling, mice were treated topically with 0.1 g of the corresponding ointment on the wound every day in each group; administrated with 0.1 g of vaseline in the control group, administrated with 0.1 g of "Jing wan hong" ointment in the positive drug group, and were not treated in model group. The treatment was administered once a day for 21 successive days (**Table 1**). All mice were humanely sacrificed by the inhalation of $CO_2$ at a gradually increasing rate of 30–70% of the chamber volume/min upon meeting the following criteria: >20% weight loss; dyspnea; and dramatic drop in body temperature. The absence of movement and breathing, as well as cardiac arrest and pupil dilation for 5 min, were used to confirm death.

**2.7.2 Measurement of wound healing rate.** The drug was administered at the same time every day, and characteristics of the wound, including color, secretions, hardness and swelling, were observed and recorded. Photos were taken on the 1st, 5th, 8th, 12th, 15th, 18th and 21st days after modeling. Then, ImageJ software was used to calculate the wound area and the rate of wound contracture as a percentage according to the following formula: wound contracture percentage on a certain day = wound area on a certain day/initial wound area × 100%.

**2.7.3 Histopathological examination.** On the 7th, 14th, and 21st days after modeling, mice in each group were anesthetized by an intraperitoneal injection of 1% pentobarbital sodium (80 mg/kg). After stripping the wound to the muscle layer, the mice were sacrificed. Take

**Table 1. Animal grouping and treatment.**

| Grouping | Dose of drug | Treated time |
|---|---|---|
| PE group | 0.1 g petroleum extract | consecutive 21 days |
| EA group | 0.1 g ethyl acetate extract | consecutive 21 days |
| NB group | 0.1 g n-butanol extract | consecutive 21 days |
| C group | 0.1 g vaseline | consecutive 21 days |
| M group | no drug administrated | consecutive 21 days |
| PD group | 0.1 g "Jing wan hong" ointment | consecutive 21 days |

Note: PE group presents petroleum extract treated group, EA group presents ethyl acetate extract treated group, NB group presents n-butanol extract treated group, C group presents control group, M group presents model group, PD group presents positive drug group.

approximately 1 mL of blood from the orbit of mice, deposit for 20 min, then centrifuge at 3000 rpm for 5 min, and take the upper serum. The upper serum layer was removed, placed in a clean centrifuge tube, and stored at -80˚C for later use. Wound tissue samples were fixed with 4% paraformaldehyde fixing solution. Paraffin sections were created, and the sections were stained with hematoxylin-eosin (HE). After staining, the samples were examined for sweat glands, inflammatory cell infiltration, fibrous tissue hyperplasia, tissue deformation, necrosis and calcification. The histopathological evaluation score was determined according to the extent of the burn and scald injuries with the following grading system: 0 points for no or minimal pathological change, 1 point for mild condition, 2 points for moderate condition, 3 points for severe condition, and 4 points for extremely severe condition.

**2.7.4 ELISA.** Mouse blood samples were removed from storage, allowed to naturally coagulate at room temperature and then centrifuged at 3000 rpm for 20 min; then, the cell supernatant was collected using sterile EP tubes. Enzyme-linked immunosorbent assay (ELISA) kits were used to measure the expression of tumor necrosis factor (TNF)-α, interleukin (IL)-10, vascular endothelial growth factor (VEGF) and transforming growth factor (TGF)-$\beta$1 according to the manufacturer's instructions.

## 2.8 Data analysis

The wound healing area data were statistically analyzed using Excel 2010 and SPSS 20.0, and experimental data from each group are expressed as the mean ± standard deviation ($\bar{x} \pm s$). P <0.05 and P <0.01 were considered statistically significant.

## 2.9 Ethics statement

Animal treatment and maintenance procedures were performed strictly in accordance with the Principle of Laboratory Animal Care and approved by the Animal Research Committee of Chongqing Three Gorges Medical College, Chongqing, China (License No. 2020–007).

## 3. Results

### 3.1 Chemical composition categories for each extraction fraction

The fraction of *Nanocnide lobata* obtained by petroleum ether extraction may contain volatile oils or grease, steroids, and triterpenoids (**Table 2**); the fraction obtained by ethyl acetate extraction may contain coumarins, lactones, and phenolic tannins (**Table 3**); the fraction obtained by n-butanol extraction may contain fragrance legumes, lactones, phenolic tannins, steroids and triterpenoids (**Table 4**); and the fraction obtained by water extraction may contain reducing sugars, polysaccharides, glycosides, phenolic tannins, amino acids, polypeptides, proteins, and organic acid class compounds (**Table 5**).

### 3.2 UPLC–MS results

Typical total ion chromatograms of the *Nanocnide lobata* samples in positive ion mode (**Fig 1A**) and negative ion mode (**Fig 1B**) are illustrated. A total of 39 compounds, including D-proline, raffinose, guanine, uridine, nicotinamide, pantothenic acid, protocatechuic acid, gentisic acid, kynurenic acid, esculin, ferulic acid, vicenin II, corymboside, isoschaftoside, caffeic acid, esculetin, cynaroside, isoorientin, N-acetyl-leucine, vitexin rhamnoside, isovitexin, isovitexin 2"-O-arabinoside, kaempferitrin, vitexin, isoscoparin-2"-beta-D-glucopyranoside, 4-hydroxy-cinnamic acid, syringaldehyde, scopoletin, salicylic acid, 2-indolecarboxylic acid, acuminoside, (+)-abscisic acid, 3-cresotinic acid, cumic alcohol, ethyl caffeate, diosmetin, skimmianine, 2-(3-methoxy-4-hydroxyphenyl)-5-(3,4-dimethoxyphenyl)-3,4-dimethyltetrahydrofur and

**Table 2. Identification test of chemical composition of petroleum ether extraction part on *Nanocnide lobata*.**

| Chemical composition | Name of experiment | Positive reaction index | Result | Conclusion |
|---|---|---|---|---|
| Volatile oil or grease | Oil spot test Phosphomolybdic acid test | The oil spot can volatilize without leaving a trace at room temperature, indicating that there is volatile oil. The oil spots do not disappear, indicating that there are oils and fats. | Oil spots do not disappear | √ |
| | | Yellow-green background with spots in blue | Spots are blue | √ |
| | Vanillin-sulfuric acid experiment | Spots are red, blue, purple, etc. | Spots are green | × |
| Steroids or triterpenoids | Chloroform-concentrated sulfuric acid test | The chloroform layer is red or cyan, and the sulfuric acid layer has green fluorescence when observed under UV light | The chloroform layer is red, and the sulfuric acid layer has green fluorescence | √ |
| | Acetic anhydride-concentrated sulfuric acid test | The color of the reaction solution changes from yellow→red→purple→blue→dirty green | The color of the reaction solution changes from yellow to red | √ |

Note: "√" indicates positive reaction, "×" indicates negative reaction.

The experimental results show that the petroleum ether extraction part of *Nanocnide lobata* may contain volatile oil or grease, steroids or triterpenoids.

**Table 3. Identification test of chemical composition of ethyl acetate extraction part on *Nanocnide lobata*.**

| Chemical composition | Name of experiment | Positive reaction index | Result | Conclusion |
|---|---|---|---|---|
| Flavonoids | Hydrochloric acid-magnesium powder reaction | The reaction solution or the resulting foam is red to purple | The foam is white | × |
| | Aluminum trichloride reaction | Visible yellow or yellow-green fluorescence when viewed under UV light | Spots are green | √ |
| | Ammonia fumigation test | Spots are yellow fluorescent under UV light | Spots are green | × |
| Coumarin or lactones | Fluorescence experiment | The spots were blue fluorescent when inspected under UV light, and the spots changed from fluorescent color to yellow-green after spraying 1% potassium hydroxide reagent | Spots are yellow-green | √ |
| | Iron hydroxamate reaction | The reaction solution appears orange-red or purple | The reaction solution appears orange-red | √ |
| Phenolic tannins | Ferric chloride reaction | Spots are green, blue-green, dark green, bluish-purple | Spots are blue | √ |
| | Gelatin test Vanillin hydrochloric acid reaction | Precipitation is produced Spots appear red to varying degrees | No precipitation is formed Spots are green | × × |
| Anthraquinones | lye test | The reaction solution is red, add hydrogen peroxide to heat, the red does not disappear, acidify with hydrochloric acid, the red disappears | The reaction solution has no color change | × |
| | Boric acid solution test | Spots are orange-yellow or red and fluoresce when viewed under UV light | No change in spots, no fluorescent color | × |
| Alkaloids | Silicotungstic acid test | Light yellow or gray white precipitates fromed | No precipitate was formed, the solution was yellow | × |
| | Bismuth potassium iodide test | A yellow or orange-red precipitate formed | No precipitate formed, the solution was brown | × |
| | Potassium iodide test | A brown precipitate formed | No precipitate was formed, the solution was yellow-brown | × |

Note: "√" indicates positive reaction, "×" indicates negative reaction.

The experimental results show that the ethyl acetate extraction part of *Nanocnide lobata* may contain coumarin or lactones and phenolic tannins.

**Table 4. Identification test of chemical composition of n-butanol extraction part on *Nanocnide lobata*.**

| Chemical composition | Name of experiment | Positive reaction index | Result | Conclusion |
|---|---|---|---|---|
| Flavonoids | Hydrochloric acid magnesium powder test | The reaction solution or the generated foam shows purplish red | White foam formed | × |
| | Aluminum trichloride reaction | Visible yellow or yellow-green fluorescence when viewed under UV light | No visible color change | × |
| | Ammonia fumigation test | Spots are yellow fluorescent under UV light | No visible color change | × |
| Coumarin or lactones | Fluorescence experiment | The spots are blue fluorescent when inspected under UV light, and the fluorescent color of the spots changed to yellow-green after spraying with 1% potassium hydroxide reagent | Spots are yellow-green | √ |
| | Iron hydroxamate reaction | The reaction solution is orange-red or purple-red | The reaction solution is orange-red | √ |
| Anthraquinones | lye test | The reaction solution is red, add hydrogen peroxide to heat, the red does not disappear, acidify with hydrochloric acid, the red disappears | No visible color change | × |
| | Boric acid solution test | Spots are orange-yellow or red and fluoresce when viewed under UV light | No change in spots, no fluorescent color | × |
| Phenolic tannins | Gelatin test | Precipitation formed | No precipitation formed | × |
| | Vanillin hydrochloric acid reaction | Spots appear red to varying degrees | Spots are colorless | × |
| | Ferric chloride reaction | Spots are green, blue-green, dark green, blue-purple | Spots are yellow-green | √ |
| Steroids or triterpenoids | Chloroform-concentrated sulfuric acid test | The chloroform layer is red or cyan, and the sulfuric acid layer has green fluorescence when observed under UV light | The chloroform layer is red, and the sulfuric acid layer has green fluorescence | √ |
| | Acetic anhydride-concentrated sulfuric acid test | The color of the reaction solution changes from yellow→red→purple→blue→dirty green | The color of the reaction solution changes from yellow to red | √ |
| Alkaloids | Silicotungstic acid test | A pale yellow or off-white precipitate formed | No precipitate was formed, the solution was yellow | × |
| | Bismuth potassium iodide test | A yellow or orange-red precipitate formed | No precipitate formed, the solution was brown | × |
| | Potassium iodide test | A brown precipitate formed | No precipitate was formed, the solution was yellow-brown | × |

Note: "√" indicates positive reaction, "×" indicates negative reaction.

The experimental results show that the n-butanol extraction part of *Nanocnide lobata* may contain coumarin or lactones, phenolic tannins, steroids or triterpenoids.

usniacin, were identified or tentatively characterized. UPLC–MS data, such as retention times, chemical formulas, compounds and main fragment ions, are summarized in **Table 6**. The above spectrum-effect relationship results illustrated that peaks P1 to P39 (**S1 Table**) were compounds in *Nanocnide lobata* potentially involved in the treatment of burn and scald injuries.

### 3.3 Observation of scald/burn wounds

By arranging the photos of wounds in each group of mice at different times with the same scale adjustment, the wound recovery in each group could be visually observed. The results are shown in **Fig 2**. From the 2[nd] day to the 8[th] day, the wounds in each group showed blisters, with thin, brownish-yellow, soft crusts. The ointments applied in each group adhered well to the wounds. From the beginning of the 2[nd] day, the wounds in the petroleum ether extract-treated group became hard, and the wounds in the control group were wet. Due to the darker color of the ointment, the wounds in the petroleum ether extract-treated group were darker. From the 8[th] day to the 12[th] day, slight effusion was observed in the control and model groups.

**Table 5. Identification test of chemical composition of water extraction part on *Nanocnide lobata*.**

| Chemical composition | Name of experiment | Positive reaction index | Result | Conclusion |
|---|---|---|---|---|
| Reducing sugars, polysaccharides and glycosides | Molich reaction | A red halo appeared between the two liquid surfaces | A red circle appears | √ |
| | Fehling reaction | Precipitation changes from light blue→brown→brick red | A brick red precipitate is formed | √ |
| Saponins | foam test | Foam appears after continuous shaking and does not disappear for a long time | A small amount of foam, which disappears after a short period of time | × |
| | Acetic anhydride-concentrated sulfuric acid test | The color of the reaction solution is from yellow→red→purple→blue→green | No significant changes | × |
| Phenolic tannins | Vanillin hydrochloric acid reaction | Spots are red | Spots are colorless | × |
| | Ferric chloride reaction | Green, dark green, bluish black or dark purple | Blue-black | √ |
| | Gelatin test | A white precipitate is formed | No precipitation is formed | × |
| Amino acids, peptides and proteins | Ninhydrin test | Blue-purple or bright yellow | No color reaction | × |
| | Biuret test | Purple or bluish-purple | No color reaction | × |
| | Heating precipitation test | Heating and boiling → turbid precipitation | Precipitation is formed | √ |
| Organic acids | Bromocresol green test | Spots are yellow on a blue background | yellow | √ |
| | Bromophenol blue test | Spots are yellow on a blue background | yellow | √ |
| | PH paper test | The color of the test paper is below PH7 | PH between 6~7 | √ |

Note: "√" indicates positive reaction, "×" indicates negative reaction.

The experimental results show that the water extraction part of *Nanocnide lobata* may contain reducing sugars, polysaccharides and glycosides, phenolic tannins, amino acids, polypeptides and proteins, organic acid compounds.

From the 12th day onward, the wounds in each group showed an obvious decrease in the degree of wound contracture, except the wounds in the model group remained larger than those in the other groups. After the 15th day, the wounds in each group had healed well, with no bleeding, and were brownish in color. The wound area was significantly smaller than that in the other groups. On the 21st day, the treatment effect in the petroleum ether extract-treated group was the most significant; the wound had basically healed, and the healed surface was smooth.

## 3.4 Wound healing rate results

After the 8th day, the wound area as a percentage of the initial area was the smallest in the positive drug group, followed by the petroleum ether extract-treated group and n-butanol extract-treated group, with a significant difference in the petroleum ether extract-treated group and positive drug group compared with the model group ($P<0.05$). After the 12th day, a significant difference was observed in the petroleum ether extract-treated group and the positive drug group compared with the model group ($P<0.01$). From the 18th day, the smallest wound area as a percentage was observed in the petroleum ether extract-treated group, followed by the control group and n-butanol extract-treated group (S2 Table). The percent remaining wound area in the ethyl acetate extract-treated group was significantly different from that in the control group. The percent remaining wound area was similar in the ethyl acetate extract-treated group and the model group and lowest in the petroleum ether extract-treated group (Table 7).

PE group presents petroleum extract treated group (0.1 g petroleum extract each day), EA group presents ethyl acetate extract treated group (0.1 g ethyl acetate extract each day), NB

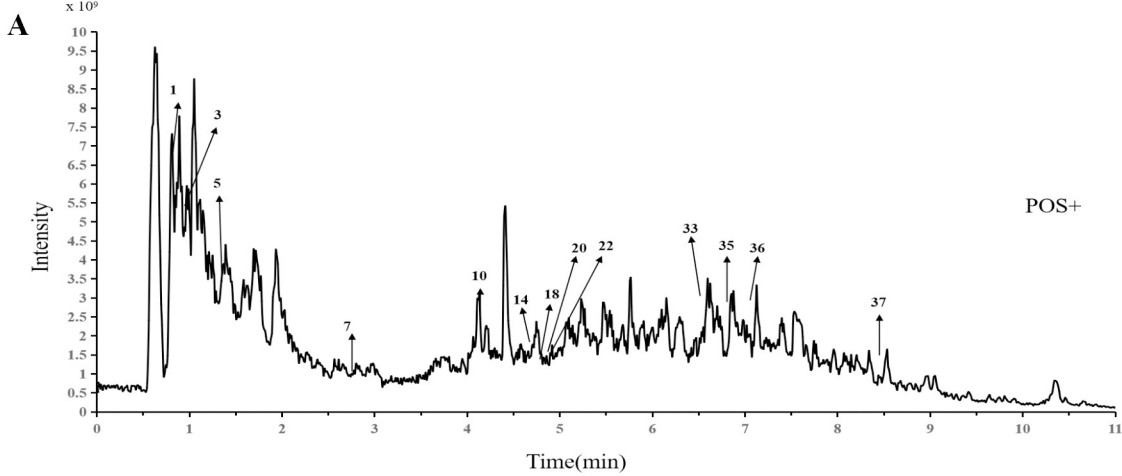

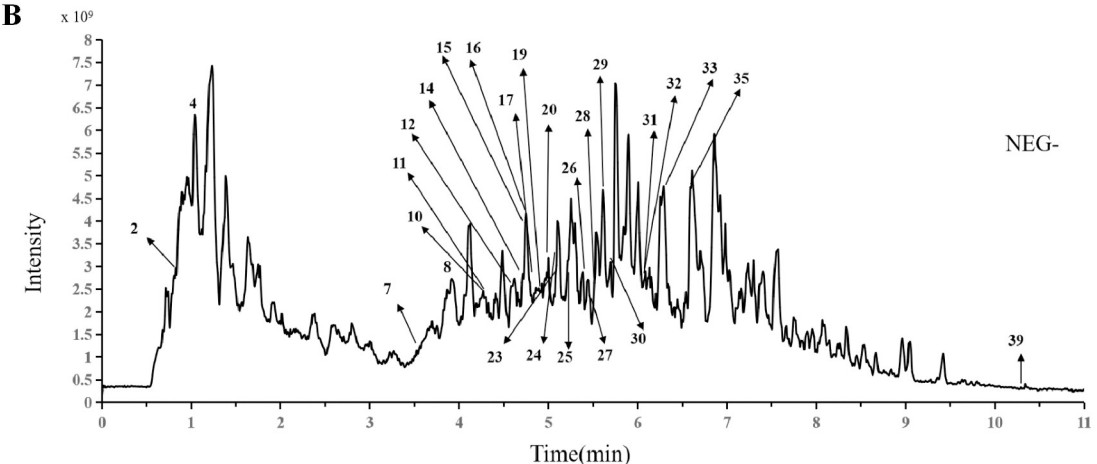

**Fig 1.** The UPLC-MS analysis of *Nanocnide lobata* extract (**A**) positive ion mode, (**B**) negative ion mode.

group presents n-butanol extract treated group (0.1 g n-butanol extract each day), C group presents control group (0.1 g vaseline each day), M group presents model group (no drug administrated), PD group presents positive drug group (0.1 g "Jing wan hong" ointment each day).

### 3.5 HE staining results

On the 7th day after modeling, no significant effect on wound healing was observed in the petroleum ether extract-treated group (**Fig 3A**) or the n-butanol extract-treated group (**Fig 3B**). Observation of the pathological sections from the mice in the petroleum ether extract-treated group (**Figs 4A and 5A**) and the n-butanol extract-treated group (**Figs 4B and 5B**) showed a gradual decrease in the number of inflammatory cells with increasing time until the 14th and 21st days after modeling. It was also observed that the bleeding was well controlled, indicating that the wounds had basically healed. In addition, in the ethyl acetate extract-treated group (**Figs 3C, 4C and 5C**), most of the necrotic epidermal tissue sloughed off, epidermal migration was observed on the surface of the wound edge, increases in the numbers of fibro-blasts and capillaries were observed, and there was still mild inflammatory cell infiltration in

**Table 6. The typical total ion chromatogram of *Nanocnide lobata* sample in positive ion mode and negative ion mode respectively.**

| NO. | Retention time (min) | Compound | Formula | [M+H,M+H-H$_2$O] | [M-H,M+FA-H] |
|---|---|---|---|---|---|
| 1 | 0.8115 | D-proline | C$_5$H$_9$NO$_2$ | 116.0707 | - |
| 2 | 0.8260 | Raffinose | C$_{18}$H$_{32}$O | - | 539.1384 |
| 3 | 0.9802 | Guanine | C$_5$H$_5$N$_5$O | 152.0564 | |
| 4 | 1.044 | Uridine | C$_9$H$_{12}$N$_2$O$_6$ | - | 243.0629 |
| 5 | 1.362 | nicotinamide | C$_6$H$_6$N$_2$O | 123.0552 | - |
| 6 | 2.799 | Pantothenic acid | C$_9$H$_{17}$NO$_5$ | 220.1175 | - |
| 7 | 3.546 | Protocatechuic acid | C$_7$H$_6$O$_4$ | - | 153.0198 |
| 8 | 3.905 | Gentisic acid | C$_7$H$_6$O$_4$ | - | 153.0198 |
| 9 | 4.161 | Kynurenic acid | C$_{10}$H$_7$NO$_3$ | 190.0494 | - |
| 10 | 4.230 | Esculin | C$_{15}$H$_{16}$O$_9$ | - | 385.0788 |
| 11 | 4.258 | ferulic acid | C$_{10}$H$_{10}$O$_4$ | - | 193.0512 |
| 12 | 4.533 | Vicenin II | C$_{27}$H$_{30}$O$_{15}$ | - | 593.1537 |
| 13 | 4.712 | Corymboside | C$_{26}$H$_{28}$O$_{14}$ | 565.1542 | - |
| 14 | 4.719 | Isoschaftoside | C$_{26}$H$_{28}$O$_{14}$ | - | 563.1440 |
| 15 | 4.737 | caffeic acid | C$_9$H$_8$O$_4$ | - | 179.0357 |
| 16 | 4.746 | Esculetin | C$_9$H$_6$O$_4$ | - | 177.0200 |
| 17 | 4.792 | Cynaroside | C$_{21}$H$_{20}$O$_{11}$ | - | 447.0944 |
| 18 | 4.804 | Isoorientin | C$_{21}$H$_{20}$O$_{11}$ | 449.1070 | - |
| 19 | 4.879 | N-Acetyl-leucine | C$_8$H$_{15}$NO$_3$ | - | 172.0986 |
| 20 | 4.964 | Vitexin rhamnoside | C$_{27}$H$_{30}$O$_{14}$ | - | 577.1590 |
| 21 | 4.966 | Isovitexin | C$_{21}$H$_{20}$O$_{10}$ | 433.1121 | |
| 22 | 4.966 | Isovitexin 2''-O-arabinoside | C$_{26}$H$_{28}$O$_{14}$ | 565.1541 | - |
| 23 | 5.078 | Kaempferitrin | C$_{27}$H$_{30}$O$_{14}$ | - | 577.1590 |
| 24 | 5.088 | Vitexin | C$_{21}$H$_{20}$O$_{10}$ | - | 431.0998 |
| 25 | 5.182 | Isoscoparin-2''-Beta-D-glucopyranoside | C$_{28}$H$_{32}$O$_{16}$ | - | 605.1532 |
| 26 | 5.392 | 4-Hydroxcinnamic acid | C$_9$H$_8$O$_3$ | - | 163.0407 |
| 27 | 5.467 | Syringaldehyde | C$_9$H$_{10}$O$_4$ | - | 181.0514 |
| 28 | 5.513 | scopoletin | C$_{10}$H$_8$O$_4$ | - | 191.0357 |
| 29 | 5.617 | Salicylic acid | C$_7$H$_6$O$_3$ | - | 137.0247 |
| 30 | 5.714 | 2-Indolecarboxylic acid | C$_9$H$_7$NO$_2$ | - | 160.0410 |
| 31 | 6.065 | Acuminoside | C$_{21}$H$_{36}$O$_{10}$ | - | 447.2248 |
| 32 | 6.066 | (+)-Abscisic acid | C$_{15}$H$_{20}$O$_4$ | - | 263.1293 |
| 33 | 6.289 | 3-Cresotinic acid | C$_8$H$_8$O$_3$ | - | 151.0405 |
| 34 | 6.579 | Cumic alcohol | C$_{10}$H$_{14}$O | 133.1010 | - |
| 35 | 6.597 | Ethyl caffeate | C$_{11}$H$_{12}$O$_4$ | - | 207.0670 |
| 36 | 6.856 | Diosmetin | C$_{16}$H$_{12}$O$_6$ | 301.0699 | - |
| 37 | 7.157 | Skimmianine | C$_{14}$H$_{13}$NO$_4$ | 260.0912 | - |
| 38 | 8.535 | 2-(3-Methoxy-4-hydroxyphenyl)-5-(3,4-dimethoxyphenyl)-3,4-dimethyltetrahydrofuran | C$_{21}$H$_{26}$O$_5$ | 341.1738 | - |
| 39 | 10.31 | Usniacin | C$_{18}$H$_{16}$O$_7$ | - | 343.0835 |

the dermis. The treatment effect was slightly better in the ethyl acetate extract-treated group than in the model group, but the wounds had still not fully healed. On the 7[th], 14[th], and 21[st] days after modeling, no inflammatory cell aggregation was observed in pathological sections from the control group (**Figs 3D, 4D and 5D**). In the model group, large ulcers containing inflammatory cells, mild inflammation and moderate bleeding in the dermis were observed,

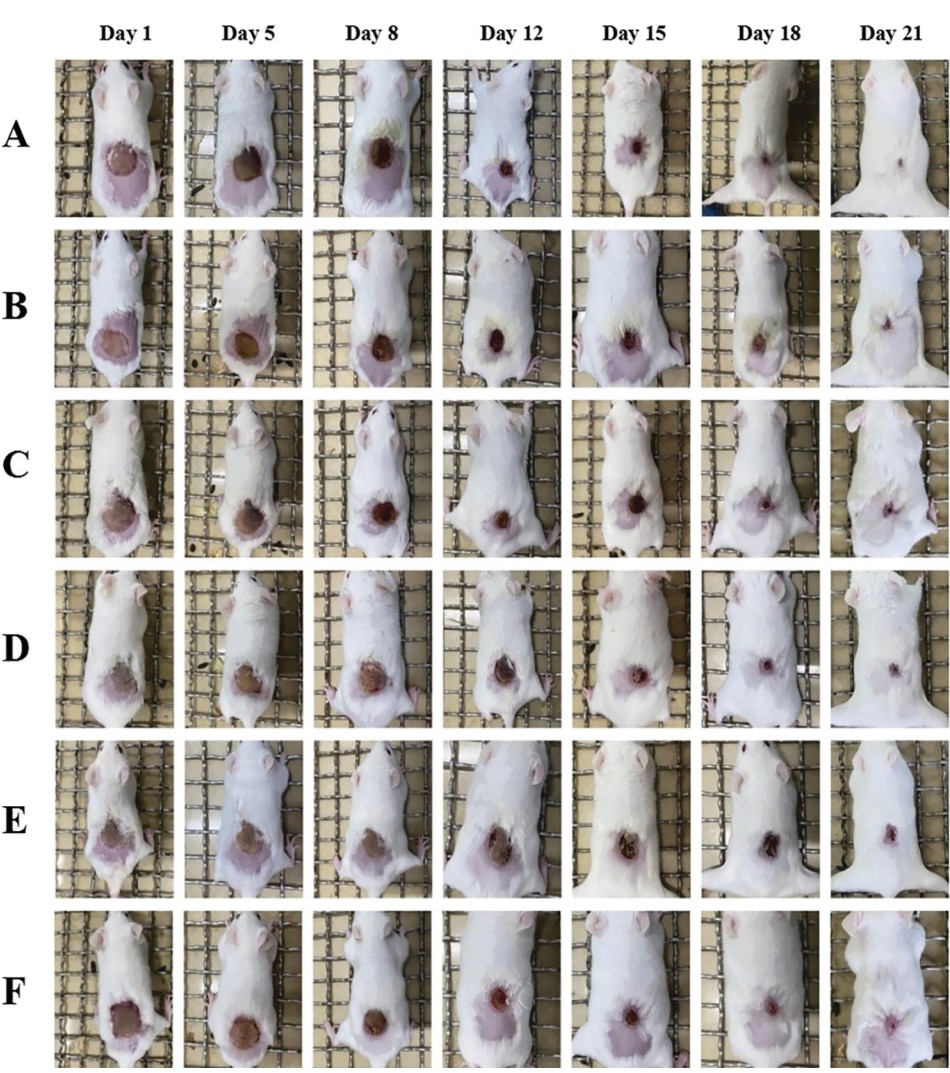

**Fig 2. General observation of the wound.**

Table 7. Change in percent contracture rate of wound ($\bar{x} \pm s$).

| Time (d) | wound contracture rate (%) | | | | | |
|---|---|---|---|---|---|---|
| | **PE group** | **EA group** | **NB group** | **C group** | **M group** | **PD group** |
| Day 1 | 100±0.0 | 100±0.0 | 100±0.0 | 100±0.0 | 100±0.0 | 100±0.0 |
| Day 5 | 83.37±11 | 99.07±11 | 84.02±29 | 96.63±24 | 92.02±25 | 73.28±17 |
| Day 8 | 70.75±17* | 75.23±23 | 64.86±34 | 95.21±6* | 78.87±26 | 50.98±4* |
| Day 12 | 20.97±7** | 53.72±20 | 37.23±21 | 58.82±18 | 59.15±18 | 18.29±10** |
| Day 15 | 15.44±6** | 33.13±13* | 36.27±16 | 34.47±13 | 47.32±15 | 15.21±7** |
| Day 18 | 3.79±4** | 11.61±8 | 20.34±24 | 5.47±17 | 20.85±16 | 7.32±4** |
| Day 21 | 0.97±1** | 4.21±7 | 4.52±6 | 1.64±6 | 11.83±8 | 1.48±2** |

Note: n = 10

*$P<0.05$

**$P<0.01$.

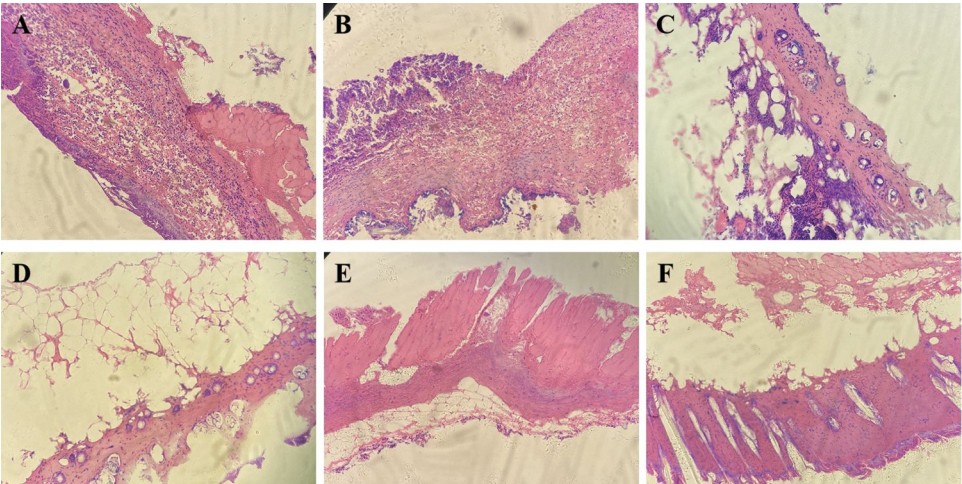

**Fig 3. Pathologic sections of wound of mice in day 7.** (**A**) petroleum ether extract-treated group, (**B**) n-butanol extract-treated group, (**C**) ethyl acetate extract-treated group, (**D**) control group, (**E**) model group, (**F**) positive drug group.

with severe peeling and crusting of the epidermis, indicating a lack of wound healing (**Figs 3E, 4E** and **5E**). Compared with those in the model group, wounds in the positive drug group showed mild epidermal hyperplasia and a large number of sweat glands generated in the dermis, indicating wound healing (**Figs 3F, 4F** and **5F**). In addition, according to the score of evaluation standards of tissues, significant pathological changes were observed (**Fig 6**). Compared with the control group, the score significant increased in the model group ($^{**}P<0.01$) while decreased in petroleum extract treated group ($^{##}P<0.01$), ethyl acetate extract group ($^{##}P<0.05$, $^{##}P<0.05$) and n-butanol extract treated group ($^{##}P<0.01$) at day 7 (**Fig 6A**), day 14 (**Fig 6B**) and day 21 (**Fig 6C**) respectively. Taken together, the histopathological results were basically consistent with the results of overall observation.

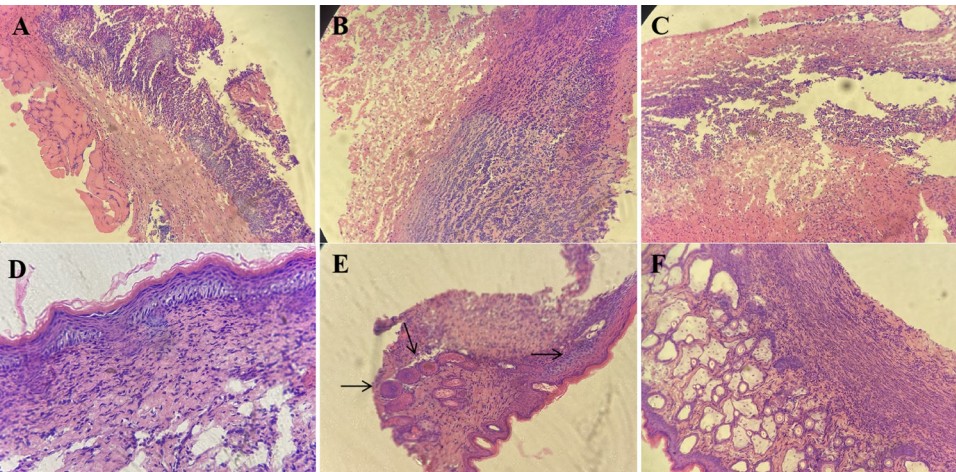

**Fig 4. Pathologic sections of wound of mice in day 14.** (**A**) petroleum ether extract-treated group, (**B**) n-butanol extract-treated group, (**C**) ethyl acetate extract-treated group, (**D**) control group, (**E**) model group, (**F**) positive drug group.

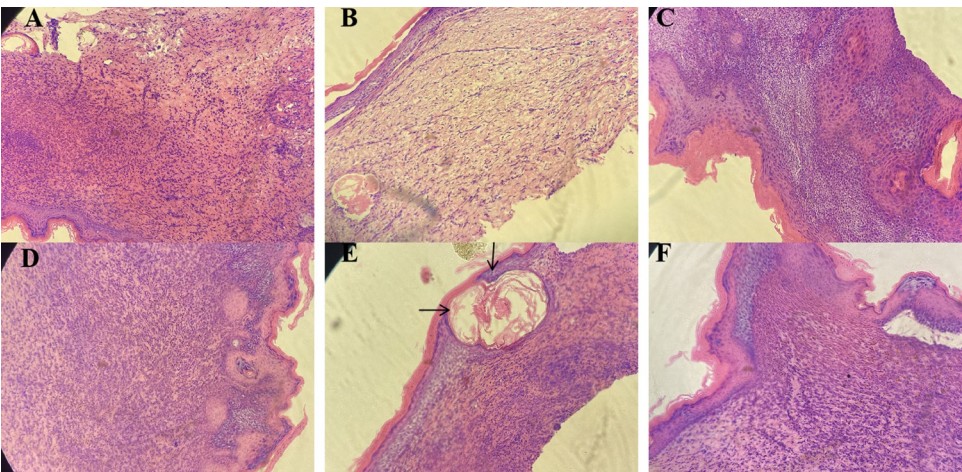

**Fig 5. Pathologic sections of wound of mice in day 21.** (**A**) petroleum ether extract-treated group, (**B**) n-butanol extract-treated group, (**C**) ethyl acetate extract-treated group, (**D**) control group, (**E**) model group, (**F**) positive drug group.

## 3.5 ELISA results

The serum levels of TNF-α and IL-10 on the 7th, 14th, and 21st days were measured to investigate the effect of *Nanocnide lobata* on the inflammatory response. The serum level of TGF-$\beta$1 was measured to study the role of *Nanocnide lobata* in promoting wound healing, and the VEGF level was measured to investigate the effect of *Nanocnide lobata* on wound contracture (S3 Table). Compared with the model group, the petroleum ether extract-treated group showed significant differences in the serum levels of TNF-α (Table 8) and IL-10 (Table 9) on the 7th, 14th, and 21st days. There was also a significant difference in the content of TGF-$\beta$1 in the petroleum ether extract-treated group on the 21st day (Table 10) and in the content of VEGF in the petroleum ether extract-treated group on the 7th and 14th days (Table 11).

## 4. Discussion

The healing burns and scalds is a natural physiological process that restores the function and integrity of damaged skin tissue [14]. The process is commonly divided into 4 overlapping stages, including coagulation, inflammation, neotissue formation and tissue remodeling. Cytokines and inflammatory mediators are involved in various processes of wound healing. The cells involved in wound healing mainly include keratinocytes, fibroblasts, endothelial cells, macrophages and platelets [15]. Cytokines include growth factors, tumor necrosis factors, interferons, colony-stimulating factors, chemokines, and interleukins [16]. The coagulation stage mainly involves platelets and fibrin, while the inflammatory stage mainly involves neutrophils, mononuclear macrophages, and lymphocytes, among others [17]. The proliferation stage mainly involves fibroblasts, endothelial cells, epithelial cells and collagen, and the remodeling stage mainly consists of collagen fiber contraction and scarring [18].

Western medicine and TCM have their own advantages and disadvantages in the treatment of burns and scalds [19]. Western medicine is well suited for treating patients with severe burns over large areas [20]. Because the onset of action is fast in Western medicine, it plays an irreplaceable role in rescuing critically burned patients. However, there are still some aspects of Western medicine that merit additional consideration, such as the long-term application of antibiotics and analgesics [21], which can lead to patients developing drug resistance and drug

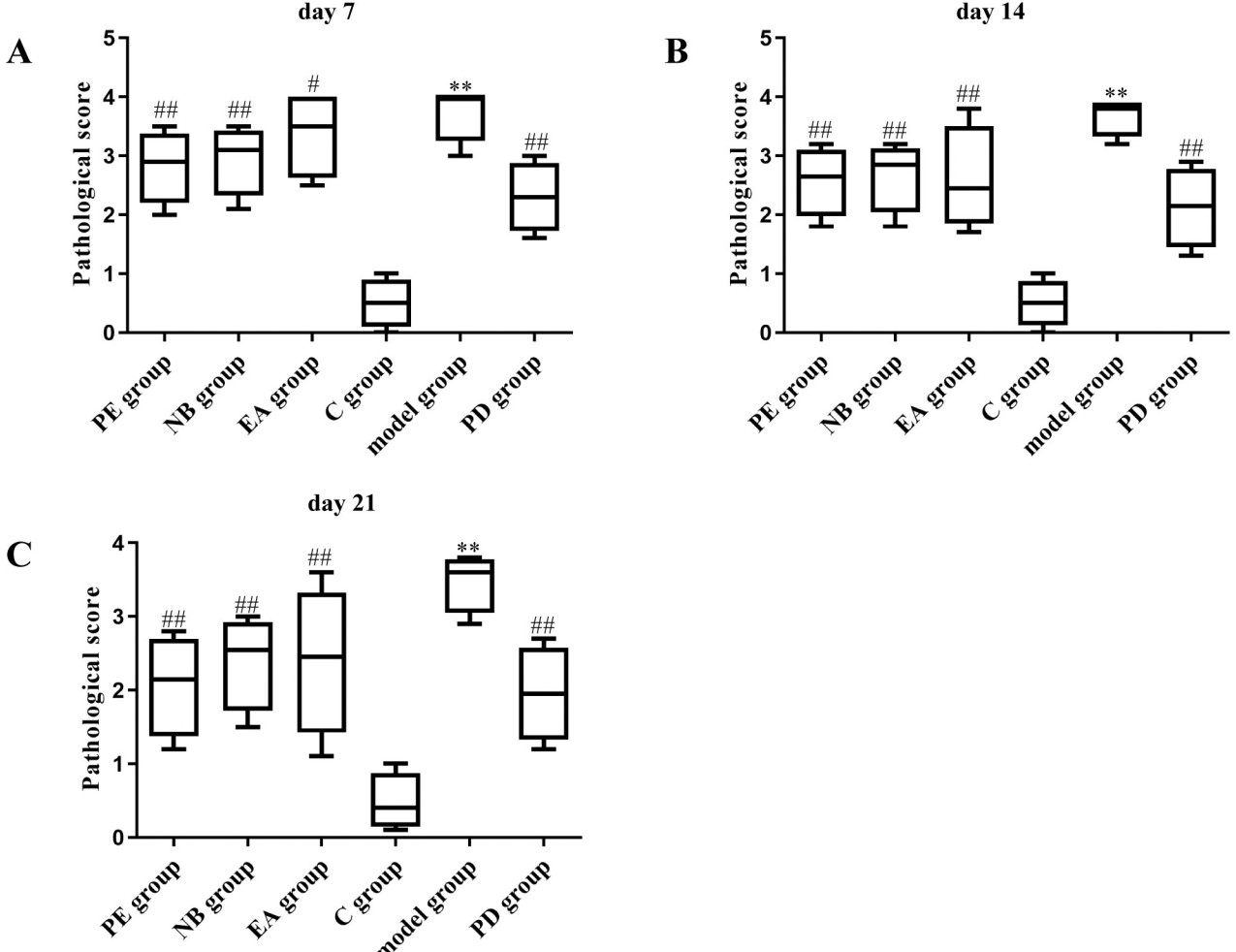

**Fig 6. The histopathological scores of the burn and scald injuries in the mice after treated with *Nanocnide lobate*.** (**A**) day 7, (**B**) day 14, (**C**) day 21.

dependence. In TCM, anti-inflammatory, muscle-building, and heat- or toxin-removing drugs are often used and are the most commonly used drugs in the treatment of burn wounds in clinical practice [22]. Moreover, most Chinese medicines contain analgesic ingredients; thus, there is no need to use additional analgesics, which avoids the development of dependence on opioid analgesics [23].

The experimental results of this study showed that the fractions of *Nanocnide lobata* extracted using petroleum ether, ethyl acetate, n-butanol and water mainly contain volatile oils, coumarins, lactones, phenolic tannins, steroids, triterpenoids, reducing sugars, polysaccharides and glycosides, phenolic tannins, amino acids, polypeptides and proteins, and organic acid compounds. Similar to our results, it has been reported that volatile oils may have a beneficial effect on the healing of burns and scalds [24,25]; reducing sugars may also be beneficial for burns and scalds [26]. In addition, phenolic tannins and phenolic compounds have been reported to have a significant curative effect on burns and scalds, which is consistent with our results [27]. Furthermore, terpenoids are considered to have anti-inflammatory and antibacterial effects in the treatment of burns and scalds [28,29]. The combination of UPLC and MS can fully leverage their advantages, significantly improving the repeatability and reliability of

**Table 8. Changes in the concentration of TNF-α in each group at different time points after injury ($\bar{x} \pm s$).**

| Group | TNF-α (pg /mL) $\bar{x} \pm s$ | | |
|---|---|---|---|
| | Day 7 | Day 14 | Day 21 |
| PE group | 161.67±4.93** | 106.33±3.21* | 77.67±4.04** |
| EA group | 194.57±5.86 | 156.38±2.08 | 98.65±2.08 |
| NB group | 191.69±6.11 | 148.31±4.16 | 99.01±1.00 |
| C group | 193.33±1.53 | 153.60±16.07 | 95.77±8.14 |
| M group | 199±1.02 | 153.39±15.27 | 103.33±2.87 |
| PD group | 157.67±1.53** | 103.03±4.25* | 78.01±2.19** |

Note: n = 10

*$P<0.05$

**$P<0.01$.

PE group presents petroleum extract treated group (0.1 g petroleum extract each day), EA group presents ethyl acetate extract treated group (0.1 g ethyl acetate extract each day), NB group presents n-butanol extract treated group (0.1 g n-butanol extract each day), C group presents control group (0.1 g vaseline each day), M group presents model group (no drug administrated), PD group presents positive drug group (0.1 g "Jing wan hong" ointment each day).

quantitative analysis, as well as the accuracy of qualitative analysis. It has well adapted to the demand for automated and high-throughput analysis methods in modern drug research and has become one of the important methods for drug analysis (**Table 6**). Therefore, bioactive compounds in *Nanocnide lobata* were identified by UPLC–MS. Among them, ferulic acid, kaempferitrin, caffeic acid, and salicylic acid have been confirmed to exhibit anti-inflammatory and antioxidant activity related to the treatment of burns and scalds. The anti-inflammatory effect of ferulic acid is mainly related to the levels of PPARγ and CAM and the NF-κB and p38 MAPK signaling pathways. Ferulic acid also plays an antifibrotic role via TGF-β/Smad and MMP/TIMP signaling [30]. A study on the anti-inflammatory activity of kaempferitrin showed that it inhibited leukocyte infiltration and exudation in mice with bradykinin-induced pleurisy [31]. Caffeic acid has been reported to exhibit anti-inflammatory, antioxidant and immunomodulatory properties by inhibiting NF-κB activation and transcriptional activity of

**Table 9. Changes in the concentration of IL-10 in each group at different time points after injury ($\bar{x} \pm s$).**

| Group | IL-10 (pg /mL) $\bar{x} \pm s$ | | |
|---|---|---|---|
| | Day 7 | Day 14 | Day 21 |
| PE group | 291.77±4.93* * | 185.09±9.54** | 141.33±1.53* * |
| EA group | 328.51±8.51 | 212.02±3.61 | 164.03±6.56 |
| NB group | 327.95±6.43 | 215.00±4.93 | 167.89±8.02 |
| C group | 323.33±1.53 | 215.33±5.51 | 173.23±2.01 |
| M group | 338.04±1.21 | 227.10±6.08 | 179.56±1.75 |
| PD group | 289.79±4.73* * | 175.69±2.52** | 146±2.66* * |

Note: n = 10

*$P<0.05$

**$P<0.01$.

PE group presents petroleum extract treated group (0.1 g petroleum extract each day), EA group presents ethyl acetate extract treated group (0.1 g ethyl acetate extract each day), NB group presents n-butanol extract treated group (0.1 g n-butanol extract each day), C group presents control group (0.1 g vaseline each day), M group presents model group (no drug administrated), PD group presents positive drug group (0.1 g "Jing wan hong" ointment each day).

**Table 10. Changes in the concentration of TGF-β1 in each group at different time points after injury ($\bar{x} \pm s$).**

| Group | TGF-β1 (pg /mL) $\bar{x} \pm s$ | | |
|---|---|---|---|
| | **Day 7** | **Day 14** | **Day 21** |
| PE group | 100.01±4.58 | 97.67±4.16 | 75.68±3.06* |
| EA group | 113.02±13.11 | 116.34±6.81 | 113.12±7.21 |
| NB group | 111.33±6.81 | 118.23±13.11 | 118.97±2.52 |
| C group | 97.23±11.00 | 108.93±9.45 | 101.52±3.06 |
| M group | 117.15±12.11 | 114.69±11.72 | 115.01±5.03 |
| PD group | 88±4.58* | 108±4.58 | 91.77±3.01* |

Note: n = 10

*$P < 0.05$.

PE group presents petroleum extract treated group (0.1 g petroleum extract each day), EA group presents ethyl acetate extract treated group (0.1 g ethyl acetate extract each day), NB group presents n-butanol extract treated group (0.1 g n-butanol extract each day), C group presents control group (0.1 g vaseline each day), M group presents model group (no drug administrated), PD group presents positive drug group (0.1 g "Jing wan hong" ointment each day).

the COX-2 gene in epithelial cells and iNOS gene expression and NO production in macrophage cell lines [32]. Salicylic acid plays a crucial role in defending against pathogenic agents and exerts anti-inflammatory effects through suppressing the transcription of cyclooxygenase genes [33]. In this mouse model of deep second-degree burns and scalds, macroscopic observation of wounds, calculation of the wound contracture and observation of pathological sections revealed the best therapeutic effect of *Nanocnide lobata* in the petroleum ether-treated group. The results of this study showed that secretions, redness and swelling of the burn and scald wounds resolved more quickly in the petroleum extract-treated group and n-butanol extract-treated group than in the model group ($P < 0.05$). Additionally, the wound healing rate was higher in the petroleum extract-treated group than in the ethyl acetate extract-treated group and n-butanol extract-treated group ($P < 0.05$), indicating a definite effect of *Nanocnide lobata* in the treatment of burns and scalds in mice. This drug may help to shorten the

**Table 11. Changes in the concentration of VEGF in each group at different time points after injury ($\bar{x} \pm s$).**

| Group | VEGF (pg /mL) $\bar{x} \pm s$ | | |
|---|---|---|---|
| | **Day 7** | **Day 14** | **Day 21** |
| PE group | 266.67±4.73* | 311.33±10.50* | 270.33±5.13 |
| EA group | 228.00±3.01 | 264.33±7.02 | 281.36±8.62 |
| NB group | 238.33±3.51 | 269.67±6.11 | 280±8.02 |
| C group | 217.33±11.59 | 235.01±8.12 | 225.01±5.02 |
| M group | 217.67±14.64 | 239.33±17.6 | 247.11±14.03 |
| PD group | 315.67±9.5** | 322.67±2.08* | 320±5.58* |

Note: n = 10

*$P < 0.05$

**$P < 0.01$.

PE group presents petroleum extract treated group (0.1 g petroleum extract each day), EA group presents ethyl acetate extract treated group (0.1 g ethyl acetate extract each day), NB group presents n-butanol extract treated group (0.1 g n-butanol extract each day), C group presents control group (0.1 g vaseline each day), M group presents model group (no drug administrated), PD group presents positive drug group (0.1 g "Jing wan hong" ointment each day).

duration of clinical symptoms and promote rapid recovery from such wounds. In the clinical use of *Nanocnide lobata* for the treatment of burns and scalds, soaking with rapeseed oil is also applied, which is consistent with the active compounds (petroleum extract-treated group) identified in this study.

TNF-α and IL-10 are representative inflammatory factors involved in the inflammatory phase [34]. TNF-α is related to the initiation of early wound healing, while IL-10 can regulate the early inflammatory response to reduce stromal activity and avoid scar formation [35]. TGF-β1 can promote wound healing, but its overexpression can increase scar formation [36]. VEGF is one of the most potent angiogenic growth factors in skin. VEGF contributes to wound contracture, and while its underexpression can lead to wound healing, its overexpression may increase scar formation [37]. Several studies have shown that many important biological factors, including TNF-α, VEGF, and TGF-1, are involved in the process of burn wound repair [38,39]. The body is prompted to release proinflammatory cytokines such as TNF-α when the skin is injured and an infection develops. The cytokine eventually starts an inflammatory cascade reaction to eliminate necrotic cells and tissues [40]. However, individuals who have excessive inflammation may develop systemic inflammatory response syndrome (SIRS) and immunological dysfunction, which can be life-threatening [41]. Anti-inflammatory medications are therefore crucial for the healing of burn wounds. In the present study, significant differences in the serum levels of TNF-α and IL-10 were observed in the petroleum ether-treated group at the 7th, 14th, and 21st days compared with the model group. There was also a significant difference in the content of TGF-β1 in the petroleum ether-treated group at the 21st day compared with the model group and in the content of VEGF in the petroleum ether-treated group at the 7th and 14th days compared with the model group. An increase and then a decrease were observed in the levels of inflammatory indicators in the petroleum ether-treated group, with significant differences on the 7th and 14th days compared with the model group.

## 5. Conclusion

Taken together, the results of this study revealed that the main types of components in *Nanocnide lobata* include volatile oils, coumarins, and lactones. Among them, ferulic acid, kaempferitrin, caffeic acid, and salicylic acid may exhibit beneficial activities for the healing of burns and scalds. The ointment used in the petroleum ether treated group and the volatile oil compounds of *Nanocnide lobata* might be effective drugs in the treatment of burns and scalds, with protective pharmacological effects achieved by reducing the expression of TNF-α, IL-10 and TGF-β1 and increasing the expression of VEGF, in turn promoting wound tissue repair, accelerating wound healing, exerting anti-inflammatory and analgesic effects, and even reducing scar tissue proliferation.

## Supporting information

**S1 Table. The detailed information of components identified in *Nanocnide lobata* based on UPLC-MS.**
(XLSX)

**S2 Table. The detailed information of contracture rate of wound.**
(XLSX)

**S3 Table. The detailed information of ELISA on TNF-α, IL-10, TGF-β1, VEGF respectively.**
(XLSX)

## Acknowledgments

All of the authors have developed research plans and participated in research design, manuscript development, editing, and completion of manuscripts. All authors contributed to manuscript revision, read and approved the submitted version.

## Author Contributions

**Conceptualization:** Dongyang Yi, Jingxin Mao.

**Data curation:** Yanlin Zou, Qian Huang, Jingxin Mao.

**Formal analysis:** Qian Huang, Xiaolong Zhu, Jingxin Mao.

**Funding acquisition:** Yanlin Zou.

**Investigation:** Yanlin Zou, Cao Yu.

**Methodology:** Yanlin Zou, Cao Yu, Qian Huang, Xiaoyan Tan.

**Project administration:** Cao Yu, Qian Huang, Xiaorong Tan, Dongyang Yi.

**Resources:** Xiaoyan Tan, Xiaolong Zhu.

**Software:** Xiaorong Tan, Xiaoyan Tan, Jingxin Mao.

**Supervision:** Xiaorong Tan.

**Visualization:** Yanlin Zou.

**Writing – original draft:** Yanlin Zou.

**Writing – review & editing:** Dongyang Yi, Jingxin Mao.

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
