## [Decision Letter · Decision Letter 0]

14 Feb 2023

PONE-D-22-27583Investigating active chemical constituents and pharmacology of Nanocnide lobata  in the treatment of burn and scaldPLOS ONE

Dear Dr. mao,

Thank you for submitting your manuscript to PLOS ONE. After careful consideration, we feel that it has merit but does not fully meet PLOS ONE’s publication criteria as it currently stands. Therefore, we invite you to submit a revised version of the manuscript that addresses the points raised during the review process.

We look forward to receiving your revised manuscript.

Kind regards,

Sairah Hafeez Kamran, PhD

Academic Editor

PLOS ONE

Journal Requirements:

“This work was supported by Chongqing Municipal Education Commission Science and Technology Research Project---“Research on the active ingredients of Nanocnide lobata (snow medicine) in the treatment of burns and scald” (NO. KJQN201802705); Chongqing Municipal Health and Family Planning Commission Traditional Chinese Medicine Science and Technology Project---“Pharmacodynamic observation and mechanism study of Nanocnide lobata (snow medicine) on burns and scald” (NO. ZY201702136) respectively.”

4**. **Your ethics statement should only appear in the Methods section of your manuscript. If your ethics statement is written in any section besides the Methods, please move it to the Methods section and delete it from any other section. Please ensure that your ethics statement is included in your manuscript, as the ethics statement entered into the online submission form will not be published alongside your manuscript.

Additional Editor Comments (if provided):

Dear Authors

The manuscript requires scientific editing. The active plant extracts require quantitative analysis. The histopathological analysis also require detailed elucidation with proper marking in the figures of the prominent features

Reviewers' comments:

Reviewer's Responses to Questions

**Comments to the Author**

1. Is the manuscript technically sound, and do the data support the conclusions?

Reviewer #1: Partly

Reviewer #2: Yes

2. Has the statistical analysis been performed appropriately and rigorously? 

Reviewer #1: Yes

Reviewer #2: Yes

3. Have the authors made all data underlying the findings in their manuscript fully available?

Reviewer #1: Yes

Reviewer #2: Yes

4. Is the manuscript presented in an intelligible fashion and written in standard English?

Reviewer #1: Yes

Reviewer #2: No

5. Review Comments to the Author

Reviewer #1: 1. There should be more investigation of chemical composition of different extracts that used in this Study and it is unacceptable to say 'may contains' that repeated for all extracts. You should use advanced method of analysis like GC- MS or HPLC or HPLC MS to be sure about the chemical constituents of the used extracts.

2. What is the time of plant collection?

3. In line 80: Using 15% of the plant extracts. What is your reference about that?

4. Resolution of picture needs to be improved.

5. Tables 5-9: Groups should be rewritten with clarified way.

Reviewer #2: Reviewer Comments:

1. Authors should rectify the grammar mistake throughout the manuscript.

2. Abstract:

Authors frequently used abbreviations in the whole abstract section.

Line#7: Confirm the “colorrections” spelling.

3. Introduction:

Line # 57 to 61, Please rewrite the whole sentences because the meaning of these is not clear.

4. Methodology section:

Line # 88 to 94, the authors should use a small letter throughout a sentence.

In lines # 95 to 97, the authors should use an ELISA test kit.

In line # 94, kindly check, whether the sentence is in the present tense or past tense.

In line # 100, authors should use the first full form then next time use abbreviations.

5. Result section:

In lines # 182 to 187, the authors should mention Table-1 with these lines.

In lines # 216 to 221, the authors should rectify the typographic and spelling mistakes.

6. Discussion section:

In lines # 254 to 256, the authors should complete the sentence (especially neotissue formation and tissue remodeling-----).

In lines # 256 to 259, rephrase the sentence.

In line # 274, explain the local burn wound. The authors should remove the word “local”.

In line # 282, rectify the “revelaed” spellings and check spelling throughout the manuscript.

In lines # 298 to 299, mention which active part.

Authors may relate their research work with existing literature.

7. Conclusion: the conclusion section should be concise.

6. PLOS authors have the option to publish the peer review history of their article (what does this mean?). If published, this will include your full peer review and any attached files.

Reviewer #1: No

Reviewer #2: **Yes: **Mehwish Mushtaq

---

## [Author Response · Author response to Decision Letter 0]

31 Mar 2023

Review Comments to the Author

Dear editor and reviewer,

Thanks for your letter and for the reviewer and editor’ comments concerning our manuscript entitled "Investigating active chemical constituents and pharmacology of Nanocnide lobata in the treatment of burn and scald" (Manuscript ID PONE-D-22-27583). We really appreciate Plos One for giving us an opportunity to revise our manuscript. The manuscript has been carefully and extensively revised according to the editor and the reviewers’ helpful comments. The changes have been highlighted in blue (reviewer 1#), red (reviewer 2#) and green (editor) of the revised manuscript respectively, all the co-authhave read and approved the revised manuscript.

And the responses were as follows:

Reviewer #1: 

1. There should be more investigation of chemical composition of different extracts that used in this study and it is unacceptable to say 'may contains' that repeated for all extracts. You should use advanced method of analysis like GC- MS or HPLC or HPLC MS to be sure about the chemical constituents of the used extracts.

Response: Following your suggestion, UPLC-MS experiment has been carried out for clarify the chemical constituents of the used extracts. Please check the details of results in figure 1 and Table 5.

2.What is the time of plant collection?

Response: The time of plant collection is April of 2021 which has been added in the manuscript.

3.In line 80: Using 15% of the plant extracts. What is your reference about that?

Response: I’m so sorry for the mistake. The sentence of the meaning is “The above 3 parts are mixed evenly with the proportion of 85% white vaseline and 15% extractions (petroleum ether extraction, ethyl acetate ester extraction, and n-butanol extraction) respectively. For the purpose to obtain an ointment of suitable consistency for the screening experiment of active fractions for the further study.” which has been revised in line 81 to line 85.

4.Resolution of picture needs to be improved.

Response: Thanks for your careful work. The images of whole manuscript has been improved and re-uploaded as separate files.

5. Tables 5-9: Groups should be rewritten with clarified way.

Response: Thank you for your help. The groups have been rewritten with clarified and briefly way. Please check in the table captions part of the manuscript.

Reviewer #2: Reviewer Comments:

1.Authors should rectify the grammar mistake throughout the manuscript.

Response: Thank you for your kindly suggestion. The professor Wang of Chongqing Medical and Pharmaceutical College was invited to revised the manuscript. Please check the modified part which highlight in red.

2. Abstract:

Authors frequently used abbreviations in the whole abstract section.

Response: Due to the limitation of the magazine on the number of words in the abstract, the author used a lot of abbreviations in the abstract section. Please kindly be informed. Following your advice, the abbreviations was improved. Please check it in line 13 to line 17.

Line#7: Confirm the “colorrections” spelling.

Response: Thanks for your careful work. “colorrections” was revised to“color rections” . Please check it in line 7.

3. Introduction:

Line # 57 to 61, Please rewrite the whole sentences because the meaning of these is not clear.

Response: We really appreciated your careful work. Following your useful advice. The whole sentences has been revised. Please check it in line 61 to line 65.

4. Methodology section:

Line # 88 to 94, the authors should use a small letter throughout a sentence.

Response: The small letter was used throughout the sentence. 

In lines # 95 to 97, the authors should use an ELISA test kit.

Response: The enzyme linked immunosorbent assay (ELISA) test kit was inserted in the manuscript.

In line # 94, kindly check, whether the sentence is in the present tense or past tense.

Response: Thanks for your careful work. The sentence is in past tense which had been revised.

In line # 100, authors should use the first full form then next time use abbreviations.

Response: Following your advice, the issues had been solved in the whole manuscript. Please check it in line 105.

5. Result section:

In lines # 182 to 187, the authors should mention Table-1 with these lines.

Response: Following your suggestion, the Tables with these lines were provided in the manuscript. Please check it in line 224 to line 230.

In lines # 216 to 221, the authors should rectify the typographic and spelling mistakes.

Response: Thanks for your careful work. The sentence lines 280 to 286 had been rectified one by one. Please check it in line 280 to line 286.

6. Discussion section:

In lines # 254 to 256, the authors should complete the sentence (especially neotissue formation and tissue remodeling-----).

Response: Thanks for your careful work. The sentence was revised which can be easily understood. Please check it in line 313 to line 315.

In lines # 256 to 259, rephrase the sentence.

Response: Following your advice, the sentence was rephrased. Please check it in line 315 to line 317.

In line # 274, explain the local burn wound. The authors should remove the word “local”.

Response: The word “local” was removed.

In line # 282, rectify the “revelaed” spellings and check spelling throughout the manuscript.

Response: The word “revelaed” spellings was rectified to revealed. Please check it in line 341.

In lines # 298 to 299, mention which active part.

Response: The active part (petroleum ether extraction) had been mentioned. Please check it in line 359.

Authors may relate their research work with existing literature.

Response: Thanks for your kindly help. Please check the related research work (line 367 to line 375) with refs (line 515 to line 528).

7.Conclusion: the conclusion section should be concise.

Response: Following your advice, the conclusion section was rewrite which is concise and brief. Please check it in line 384 to line 392.

*Corresponding author: Jingxin Mao, Ph. D

College of Pharmaceutical Sciences, Southwest University, 400715, Chongqing, China.

School of Pharmacy, Chongqing Medical and Pharmaceutical College, 401331, Chongqing, China.

Email: mmm518@163.com or maomao1985@email.swu.edu.cn

Tel: +86-13752922258

---

## [Editor Report · Decision Letter 1]

24 Apr 2023

PONE-D-22-27583R1Investigating active chemical constituents and pharmacology of Nanocnide lobata  in the treatment of burn and scaldPLOS ONE

Dear Dr. mao,

Thank you for submitting your manuscript to PLOS ONE. After careful consideration, we feel that it has merit but does not fully meet PLOS ONE’s publication criteria as it currently stands. Therefore, we invite you to submit a revised version of the manuscript that addresses the points raised during the review process.

ACADEMIC EDITOR:

Dear Authors

As per the reviewers’ comments all amendments have been addressed. The results are well established, but the entire manuscript requires improvements and clarity in English and Scientific language.

Abstract: The section is lacking appropriate scientific language for e.g.The authors have stated “Objective: To analyze the initial chemical constituents of Nanocnide lobata and figure out the effect and active fraction in the treatment of burn and scald”.

What do the authors mean by initial chemical constituents (are they analyzing primary metabolites?) and “figure out the effect” which effect?

I believe that the Objective statement may be clearer and more understandable and what I perceive from the study, more likely may be stated as “To analyze the most effective fraction of Nanocnide lobata in the treatment of burn and scald and determine its bioactive constituents”

In the methods authors have stated paper assays and test tube assays. I believe that these are not scientific names of the assays, so please add clarity. “Petroleum extraction group” doesn’t depict the proper meaning. “Petroleum extract treated group” is one suggestion or authors may think of a better scientific depiction. What treatment was provided to the control group, model group and positive model group.

Which fraction of Nanocnide lobata was subjected to UPLC-MS analysis?

Conclusion shall also be restructured. “This study preliminarily (what do the authors mean by preliminary) identified the type of the chemical compounds and the effect of Nanocnide lobata in treatment of burn and scald, which provides
the basically
foundation for further research on its medicinal value”. This statement doesn’t provide a conclusion of the study.

The entire abstract section requires English and scientific language recheck. Many parts are vague and some are mentioned above.

Similarly Introduction section also requires a thorough recheck

Line 45: "Long-term fever is prone to various complications and life-threatening" incomplete sentence

Line 53-54: Please recheck

Line 56-57. If clinical efficacy has been proven then what are the authors trying to address in the study?

Please recheck Line 65-67.

Line 77 -85 are still not clear. The authors have mixed all the fractions and prepared an ointment, then how the authors are creating separate groups of animals with labels “Petroleum extraction group”……..

The preparation of ointment may be moved to methods section. Line 134, 137, 140, 143 headings may be restructured. Also provide references for the phytochemical tests performed for every fraction.

Line 148: “Preparation of samples” heading may be more appropriate.

Section 3.3 also requires restructuring.

Line 231 and 301, reconsider the headings

Discussion: Lines 328, 331-337, 342-348, 360-362, 378-385 require special attention as the scientific and English language are not correct.

Have the authors identified any bioactive compounds from the UPLC-MS, effective in the treatment of burns and scalds. They may mention them in the discussion section.

Conclusion section also requires restructuring.

Please recheck the table and figure numbers within text are mentioned appropriately.

We look forward to receiving your revised manuscript.

Kind regards,

Sairah Hafeez Kamran, PhD

Academic Editor

PLOS ONE

---

## [Author Response · Author response to Decision Letter 1]

10 May 2023

As per the reviewers’ comments all amendments have been addressed. The results are well established, but the entire manuscript requires improvements and clarity in English and Scientific language.

Response: Following your suggestion, the entire manuscript has been revised by American Journal Experts (https://www.aje.cn/). Please find the Editing Certificate uploaded as an attachment.

Abstract: The section is lacking appropriate scientific language for e.g.The authors have stated “Objective: To analyze the initial chemical constituents of Nanocnide lobata and figure out the effect and active fraction in the treatment of burn and scald”.

Response: Thank you for your suggestions. The sentence has been revised to “To identify the most effective fraction of Nanocnide lobata in the treatment of burn and scald injuries and determine its bioactive constituents”. Please see lines 2-3 in the manuscript.

What do the authors mean by initial chemical constituents (are they analyzing primary metabolites?) and “figure out the effect” which effect?

Response: Thank you for your comments. The authors used two methods to determine the chemical constituents of Nanocnide lobata. The basic chemical identification methods applied included a variety of color or precipitation reactions using indicator and chromogenic agents. In addition, UPLC‒MS analyses were utilized to determine specific compounds in Nanocnide lobata. The phrase “figure out the effect” was inappropriate and has been deleted from the manuscript. Please see lines 2-3 in the manuscript.

I believe that the Objective statement may be clearer and more understandable and what I perceive from the study, more likely may be stated as “To analyze the most effective fraction of Nanocnide lobata in the treatment of burn and scald and determine its bioactive constituents”

Response: Thank you for your helpful suggestion. Following your advice, the text has been revised to “To identify the most effective fraction of Nanocnide lobata in the treatment of burn and scald injuries and determine its bioactive constituents”. Please see lines 2-3 in the manuscript.

In the methods authors have stated paper assays and test tube assays. I believe that these are not scientific names of the assays, so please add clarity. “Petroleum extraction group” doesn’t depict the proper meaning. “Petroleum extract treated group” is one suggestion or authors may think of a better scientific depiction. What treatment was provided to the control group, model group and positive model group.

Response: Thank you for your helpful suggestion. The phrases “paper assays” and “test tube assays” were inappropriate and have been deleted from the manuscript. The text has been changed to the following, which is clear and brief: “Chemical identification methods were used to analyze solutions extracted from Nanocnide lobata using petroleum ether, ethyl acetate, n-butanol using a variety of color reactions” Please see lines 4-6 in the manuscript.

In addition, the phrase “petroleum extract-treated group” has been used instead of “petroleum extraction group” throughout the manuscript. Similar revisions to the “ethyl acetate extract-treated group” and “n-butanol extract-treated group” have also been made accordingly.

Which fraction of Nanocnide lobata was subjected to UPLC-MS analysis?

Response: UPLC‒MS analysis was carried out on the total ethanol extract of Nanocnide lobata.

Conclusion shall also be restructured. “This study preliminarily (what do the authors mean by preliminary) identified the type of the chemical compounds and the effect of Nanocnide lobata in treatment of burn and scald, which provides the basically foundation for further research on its medicinal value”. This statement doesn’t provide a conclusion of the study.

Response: The conclusion has been restructured and the indicated sentence revised. Please see lines 410-419 in the manuscript.

The entire abstract section requires English and scientific language recheck. Many parts are vague and some are mentioned above.

Response: We appreciate your helpful suggestion. According to your suggestion, the entire manuscript has been revised by American Journal Experts (https://www.aje.cn/). Please find the Editing Certificate uploaded as an attachment.

Similarly Introduction section also requires a thorough recheck

Response: The introduction has been revised accordingly.

Line 45: "Long-term fever is prone to various complications and life-threatening" incomplete sentence

Response: The indicated sentence has been revised to the following: “Additionally, long-term fever can easily lead to various complications and further endanger life” .Please see lines 50-51 in the manuscript.

Line 53-54: Please recheck

Response: The indicated sentence has been revised to the following: “Nanocnide lobata is a traditional Chinese medicine (TCM) that is usually used to treat lung heat and cough, scrofula, hemoptysis, burns and scalds, carbuncles, bruises, snakebites, and traumatic bleeding.” Please see lines 59-62 in the manuscript.

Line 56-57. If clinical efficacy has been proven then what are the authors trying to address in the study?

Response: We apologize for any misunderstanding. The indicated text has been revised to the following: “Nanocnide lobata is commonly used to treat burns and scalds in Chinese folk medicine. However, there has been little modern clinical research on the pharmacology and efficacy of Nanocnide lobata in the treatment of burn and scald injuries”. Please see lines 62-65 in the manuscript.

Please recheck Line 65-67.

Response: The indicated sentence has been revised to the following: “Chemical identification methods were utilized to investigate the chemical composition of each extracted fraction of Nanocnide lobata and provide a chemical foundation for further mechanistic research”. Please see lines 71-73 in the manuscript.

Line 77 -85 are still not clear. The authors have mixed all the fractions and prepared an ointment, then how the authors are creating separate groups of animals with labels “Petroleum extraction group”……..

Response: We apologize for any misunderstanding. The indicated sentence has been revised to the following: “Then, the petroleum ether extract, ethyl acetate ester extract, and n-butanol extract were each mixed with white Vaseline at a proportion of 85% white Vaseline and 15% extract to obtain ointments of suitable consistency for the further screening of active fractions.” Please see lines 90-93 in the manuscript.

The preparation of ointment may be moved to methods section. Line 134, 137, 140, 143 headings may be restructured. Also provide references for the phytochemical tests performed for every fraction.

Response: Thank you for your suggestion. The description of ointment preparation has been moved to the methods section. In addition, the headings on lines 134, 137, 140, and 143 have been restructured, and references for the phytochemical tests performed for every fraction have been provided.

Line 148: “Preparation of samples” heading may be more appropriate.

Response: The indicated heading has been revised to “Preparation of samples”, as suggested. Please see lines 145 in the manuscript.

Section 3.3 also requires restructuring.

Response: Section 3.3 has been restructured.

Line 231 and 301, reconsider the headings

Response: The headings have been reconsidered and revised.

Discussion: Lines 328, 331-337, 342-348, 360-362, 378-385 require special attention as the scientific and English language are not correct.

Response: Thank you for your advice. These sentences have been revised. Please see the detail of discussion part in the manuscript.

Have the authors identified any bioactive compounds from the UPLC-MS, effective in the treatment of burns and scalds. They may mention them in the discussion section.

Response: Thank you very much for your suggestion. The bioactive compounds identified by UPLC‒MS that are effective in the treatment of burns and scalds have been mentioned in the discussion section. Please see lines 356-371 in the manuscript.

Conclusion section also requires restructuring.

Response: The conclusion section has been restructured accordingly. Please see the detail of conclusion part in the manuscript.

Please recheck the table and figure numbers within text are mentioned appropriately.

Response: Thank you for your helpful suggestion. The table and figure numbers have been rechecked and revised as needed.

---

## [Decision Letter · Decision Letter 2]

24 May 2023

PONE-D-22-27583R2Investigating the active chemical constituents and pharmacology of Nanocnide lobata  in the treatment of burn and scald injuriesPLOS ONE

Dear Dr. mao,

Thank you for submitting your manuscript to PLOS ONE. After careful consideration, we feel that it has merit but does not fully meet PLOS ONE’s publication criteria as it currently stands. Therefore, we invite you to submit a revised version of the manuscript that addresses the points raised during the review process.

ACADEMIC EDITOR: Most of the comments of the reviewers have been addressed but few minor changes as suggested below shall be addressed.

We look forward to receiving your revised manuscript.

Kind regards,

Sairah Hafeez Kamran, PhD

Academic Editor

PLOS ONE

Journal Requirements:

Reviewers' comments:

Reviewer's Responses to Questions

**Comments to the Author**

1. If the authors have adequately addressed your comments raised in a previous round of review and you feel that this manuscript is now acceptable for publication, you may indicate that here to bypass the “Comments to the Author” section, enter your conflict of interest statement in the “Confidential to Editor” section, and submit your "Accept" recommendation.

Reviewer #3: (No Response)

2. Is the manuscript technically sound, and do the data support the conclusions?

Reviewer #3: Yes

3. Has the statistical analysis been performed appropriately and rigorously? 

Reviewer #3: Yes

4. Have the authors made all data underlying the findings in their manuscript fully available?

Reviewer #3: Yes

5. Is the manuscript presented in an intelligible fashion and written in standard English?

Reviewer #3: Yes

6. Review Comments to the Author

Reviewer #3: Review Report:

The manuscript entitled “Investigating the active chemical constituents and pharmacology of Nanocnide lobate in the treatment of burn and scald injuries” is an interesting study in which authors made attempts to contribute in the field.

Note: I reviewed the file named (PONE-D-22-27583_R2_Pdf) downloaded from “View Submission” and all my comments are according to the line numberings of the pdf file “PONE-D-22-27583_R2_Pdf”

Detailed Comments/Recommendations:

Title:

Title is well written and clearly indicating the theme of the study.

Abstract:

It will be more appropriate and impressive if the names of any two to three out of 39 compounds are mentioned in line 23.

In line 8: Kindly add gander of mice as well.

It will be more impressive if authors may use few numerical results i.e. (TNF-α and IL-10 or TGF-β1) in line 27 and 28.

Key Words:

Three out of five key words are already present in the title. It will be more impressive if the authors use the key words other than the words used in Title.

Introduction:

Overall the introduction is very well written.

Kindly add family name of Nanocnide lobata anywhere in line 54-56.

Materials and Methods:

Kindly mention the identification/herbarium number, which was obtained after identification of plant from a botanist for future reference at the end of line 80.

Line 84: Mention the technique used for evaporation of ethanol, involving standard operating conditions eg. Temperature, Pressure etc.

Line 116: Kindly mention the light and dark duration and standard feeding of animals as well.

Line 176: Kindly explain the study design clearly for example how much doses of extracts and reference drugs were administered to animals and by which route and for how many days. All study design should be quite clear for the reader. It will be more appropriate and facilitate the reader to understand if the authors mention the dosing of animals in in tabular format.

Line 203: Kindly mention how much blood was obtained and from which place eg. Heart etc.

Results & Discussion:

Results are well expressed.

Discussion is well written by the authors.

Line 357: It will be more appropriate if the authors mentions the importance/advantages of UPLC-MS used here in this study.

Table 6: Kindly mention the details of control, model and positive drug groups. Give descriptions to facilitate the reader. The table should be self-explanatory. Similarly, Tables 7-10 should also be self-explanatory.

Figures 3-5 are not explained. What does A, B, C, D, E, and F indicate? Histopathological scoring is missing.

The authors did not discuss the histopathological results.

Conclusion:

The conclusion is well written.

References:

The authors are advised to review all the references for the strict adherence of the uniformity of Journal’s references style.

Reference no 13. Lines 478-480: Reference is incomplete. Kindly add on full details.

Reference no 34. Lines 544-546: Reference is incomplete. Kindly add on full details.

Check the font style and font size of references as well. It should be uniform.

Concluding Remarks:

The paper needs above mentioned suggestions along with the English Language proof readings by some language expert. Then after fulfilling the corrections the paper will be acceptable.

7. PLOS authors have the option to publish the peer review history of their article (what does this mean?). If published, this will include your full peer review and any attached files.

Reviewer #3: **Yes: **Dr. Aamir Mushtaq

Department of Pharmaceutical Sciences

Government College university Lahore, Pakistan

---

## [Author Response · Author response to Decision Letter 2]

30 May 2023

Dear editor and reviewer,

Thanks for your letter and comments concerning our manuscript entitled "Investigating active chemical constituents and pharmacology of Nanocnide lobata in the treatment of burn and scald injuries" (Manuscript ID PONE-D-22-27583). We really appreciate Plos One for giving us an opportunity to revise our manuscript. The manuscript has been carefully and extensively revised according to your helpful comments. The changes have been highlighted in red in the revised manuscript point by point.

And the responses were as follows:

Detailed Comments/Recommendations:

Title: 

Title is well written and clearly indicating the theme of the study.

Response: Thank you for your help. 

Abstract:

It will be more appropriate and impressive if the names of any two to three out of 39 compounds are mentioned in line 23.

Response: Following your suggestion, the names of any two to three out of 39 compounds are mentioned in line 23 to line 25. Please check the detail in the manuscript.



In line 8: Kindly add gander of mice as well.

Response: Do the reviewer means “gender of mice”? The gender of the mice is female which had been provided in the abstract part of the manuscript. Please check in line 8.



It will be more impressive if authors may use few numerical results i.e. (TNF-α and IL-10 or TGF-β1) in line 27 and 28.

Response: Thanks for your careful work. Few numerical results were listed in line 29 to line 32.

Key Words:

Three out of five key words are already present in the title. It will be more impressive if the authors use the key words other than the words used in Title.

Response: The key words and title is partly same. However, if I can’t use the similar word in the key words part. It was inability to summarize or indicate the key points of the article well. Therefore, I suggest the key words may changed to “Nanocnide lobata, chemical composition, UPLC‒MS, preliminary test, burn and scald injuries, pharmacological effects” .Please check in line 41-42.

Introduction:

Overall the introduction is very well written.]

Response: Thank for your constructive comments.

Kindly add family name of Nanocnide lobata anywhere in line 54-56.

Response: The family name of Nanocnide lobata is Urticaceae which was added in the manuscript. Please check it in line 59 to 62.

Materials and Methods:

Kindly mention the identification/herbarium number, which was obtained after identification of plant from a botanist for future reference at the end of line 80.

Response: The original herbarium is kept in the 407 Natural medicinal chemistry Laboratory of the Scientific Research Center of Chongqing Three Gorges Medical College (No.20210503) .Please check it in line 82 to 85.



Line 84: Mention the technique used for evaporation of ethanol, involving standard operating conditions eg. Temperature, Pressure etc.

Response: Thanks for your careful work. The technique which was used for evaporation of ethanol listed in the manuscript. Please check in line 91.



Line 116: Kindly mention the light and dark duration and standard feeding of animals as well.

Response: The breeding environment was kept at 25±10℃, the relative humidity was 40% to 70%, and the light-dark cycle was 12 h. Adapt to feeding for consecutive 3 days, fasting 12 h before the experiment, and drinking water freely. Please check in line 123 to 125.

Line 176: Kindly explain the study design clearly for example how much doses of extracts and reference drugs were administered to animals and by which route and for how many days. All study design should be quite clear for the reader. It will be more appropriate and facilitate the reader to understand if the authors mention the dosing of animals in in tabular format.

Response: Following your suggestion, the the study design has been revised which is clearly and briefly. Please check in line 192 to line 196. In addition, the table 1 titled “Animal grouping and treatment” was added in the manuscript which mention the dosing of animals in in tabular format.



Line 203: Kindly mention how much blood was obtained and from which place eg. Heart etc.

Response: Take approximately 1 mL of blood from the orbit of mice, deposit for 20 minutes, then centrifuge at 3000 rpm for 5 minutes, and take the upper serum. Store the serum in a refrigerator at -80 ° C for later use. Please check in line 213 to 216.

Results & Discussion:

Results are well expressed.

Response: Thanks for your comments.



Discussion is well written by the authors.

Response: Thanks for your encouragement.



Line 357: It will be more appropriate if the authors mentions the importance/advantages of UPLC-MS used here in this study. 

Response: The importance/advantages of UPLC-MS used here in this study was presented in the manuscript. Please check it in 376 to 381.



Table 6: Kindly mention the details of control, model and positive drug groups. Give descriptions to facilitate the reader. The table should be self-explanatory. Similarly, Tables 7-10 should also be self-explanatory.

Response: Thanks for your careful work. The tables 8-11 was revised which mention the details of control, model and positive drug groups.The tables are self-explanatory.



Figures 3-5 are not explained. What does A, B, C, D, E, and F indicate? Histopathological scoring is missing. 

Response: The figures 3-5 has been explained in line 302 to 324, please check. In addition, the histopathological scoring was added in the manuscript. Please check in line 221 to 224 as well as figure 6 (new added).



The authors did not discuss the histopathological results. 

Response: The histopathological results has been discussed in the results part. Please check in line 302 to 329.



Conclusion:

The conclusion is well written.

Response: Thanks for your comments.

References:

The authors are advised to review all the references for the strict adherence of the uniformity of Journal’s references style.

Response: The references style was revised to the style which meet the standards of Plos One.

Reference no 13. Lines 478-480: Reference is incomplete. Kindly add on full details.

Response: The full details of references style was added.



Reference no 34. Lines 544-546: Reference is incomplete. Kindly add on full details.

Response: The full details of references style was added.



Check the font style and font size of references as well. It should be uniform.

Response: The references style was revised one by one which may meet the standards of Plos One.

Concluding Remarks: 

The paper needs above mentioned suggestions along with the English Language proof readings by some language expert. Then after fulfilling the corrections the paper will be acceptable. 

Response: Following your suggestion, the entire manuscript was revised by American Journal Experts (https://www.aje.cn/). Please check the English and scientific language certificate which uploaded as the attachment.

---

## [Editor Report · Decision Letter 3]

31 May 2023

Investigating the active chemical constituents and pharmacology of Nanocnide lobata  in the treatment of burn and scald injuries

PONE-D-22-27583R3

Dear Dr. mao,

We’re pleased to inform you that your manuscript has been judged scientifically suitable for publication and will be formally accepted for publication once it meets all outstanding technical requirements.

Kind regards,

Sairah Hafeez Kamran, PhD

Academic Editor

PLOS ONE

---

## [Editor Report · Acceptance letter]

5 Jun 2023

PONE-D-22-27583R3 

Investigating the active chemical constituents and pharmacology of *Nanocnide lobata* in the treatment of burn and scald injuries 

Dear Dr. Mao:

I'm pleased to inform you that your manuscript has been deemed suitable for publication in PLOS ONE. Congratulations! Your manuscript is now with our production department. 

Kind regards, 

on behalf of

Dr. Sairah Hafeez Kamran 

Academic Editor

PLOS ONE